# Unconventional interactions of the TRPV4 ion channel with beta-adrenergic receptor ligands

Miguel Benítez-Angeles[1], Emmanuel Juárez-González[1], Ariela Vergara-Jaque[2,3], Itzel Llorente[1], Gisela Rangel-Yescas[4] ⓘ, Stéphanie C Thébault[5], Marcia Hiriart[1], León D Islas[4], Tamara Rosenbaum[1] ⓘ

The transient receptor potential vanilloid 4 (TRPV4) ion channel is present in different tissues including those of the airways. This channel is activated in response to stimuli such as changes in temperature, hypoosmotic conditions, mechanical stress, and chemicals from plants, lipids, and others. TRPV4's overactivity and/or dysfunction has been associated with several diseases, such as skeletal dysplasias, neuromuscular disorders, and lung pathologies such as asthma and cardiogenic lung edema and COVID-19–related respiratory malfunction. TRPV4 antagonists and blockers have been described; nonetheless, the mechanisms involved in achieving inhibition of the channel remain scarce, and the search for safe use of these molecules in humans continues. Here, we show that the widely used bronchodilator salbutamol and other ligands of $\beta$-adrenergic receptors inhibit TRPV4's activation. We also demonstrate that inhibition of TRPV4 by salbutamol is achieved through interaction with two residues located in the outer region of the pore and that salbutamol leads to channel closing, consistent with an allosteric mechanism. Our study provides molecular insights into the mechanisms that regulate the activity of this physiopathologically important ion channel.

## Introduction

The transient receptor potential vanilloid 4 (TRPV4) is a $Ca^{2+}$-permeable non-selective cation channel, which can either be inhibited or potentiated in a $Ca^{2+}$ concentration-dependent fashion (Strotmann et al, 2000). This protein is found in several cell types and tissues (Rosenbaum et al, 2020), including epidermal keratinocyte cells (Chung et al, 2003); vascular endothelium (Fian et al, 2007; Tanaka et al, 2008; Yin et al, 2008); smooth muscle cells in the pulmonary aorta and artery, brain arteries (Earley et al, 2005; Yang et al, 2006), primary afferent sensory neurons that innervate the gastrointestinal tract (Holzer, 2011), and enterocytes and

enteroendocrine cells (Boesmans et al, 2011; Bellono et al, 2017); and epithelia of the trachea and lungs (especially in cilia of the bronchial epithelium) (Lorenzo et al, 2008), among others (Tian et al, 2004; Teilmann et al, 2005; Birder et al, 2007; Gevaert et al, 2007; Gradilone et al, 2007; Casas et al, 2008; Pan et al, 2008).

TRPV4 is regulated by several stimuli such as temperatures around 27°C (Güler et al, 2002), hypoosmotic conditions, mechanical stress (Liedtke et al, 2000; Liedtke & Friedman, 2003), plant chemicals such as bisandrographolide A from *Andrographis paniculata* (Smith et al, 2006), phorbol derivatives (i.e., 4$\alpha$-phorbol 12,13-didecanoate, 4$\alpha$PDD) (Watanabe et al, 2002), the flavonoid apigenin (Ma et al, 2012), and the synthetic agonist GSK1016790A (GSK) (Jin et al, 2011). Other agonists or activity modulators of TRPV4 include phosphatidylinositol 4,5-bisphosphate (PIP$_2$) (Garcia-Elias et al, 2015), 5,6-epoxyeicosatrienoic acid (5,6-EET) (Watanabe et al, 2003; Berna-Erro et al, 2017), and flavonoids (Ma et al, 2012; Wang et al, 2015). Inhibitors or blockers of TRPV4 are GSK3527497 (Brooks et al, 2019), GSK205 (Phan et al, 2009), and its derivatives (Kanju et al, 2016); ruthenium red (St Pierre et al, 2009) and $Gd^{3+}$ (Berrier et al, 1992; Phan et al, 2009); RN-1734 (Vincent et al, 2009) and RN-9893 (Vincent & Duncton, 2011); and the biflavone ginkgetin (Alharbi et al, 2021).

TRPV4 regulates cellular homeostasis of the intracellular calcium concentration ($[Ca^{2+}]_i$) (White et al, 2016) by participating in maintenance of the integrity of endothelial barriers (Nilius & Droogmans, 2001; Phuong et al, 2017; Pairet et al, 2018), osmoregulation (Liedtke et al, 2000; Liedtke & Friedman, 2003), nociception (Kanju et al, 2016), and control of vascular tone (Sonkusare et al, 2012), bone homeostasis (Masuyama et al, 2008), pulmonary (Bihari et al, 2017; Morgan et al, 2018) and renal (Liedtke & Friedman, 2003) functions, and itch (Tóth et al, 2014; Chen et al, 2021). TRPV4 has also been linked to several human congenital disorders, which have been grouped into skeletal dysplasias and neuromuscular disorders (Nilius & Owsianik, 2010; Grace et al, 2017). These diseases encompass progressive degeneration of peripheral nerves and lack of establishment and development of the hard skeletal tissues, and highlight the importance of TRPV4 in human pathophysiology.

[1]Departamento de Neurociencia Cognitiva, Instituto de Fisiología Celular, Universidad Nacional Autónoma de México (UNAM), México, México  [2]Center for Bioinformatics, Simulation and Modeling, Faculty of Engineering, Universidad de Talca, Talca, Chile  [3]Millennium Nucleus of Ion Channel-Associated Diseases, Santiago, Chile  [4]Departamento de Fisiología, Facultad de Medicina, UNAM, México, México  [5]Instituto de Neurobiología, UNAM, Campus UNAM-Juriquilla, Querétaro, México

Correspondence: trosenba@ifc.unam.mx

Important roles of TRPV4 in the respiratory system have also been suggested. Smooth muscle cells, fibroblasts, submucosal glands, macrophages, vascular endothelial cells, and bronchial, tracheal, and alveolar epithelia (Liedtke et al, 2000; Delany et al, 2001; Alvarez et al, 2006; Jia & Lee, 2007; Xia et al, 2018; Palaniyandi et al, 2020) express TRPV4, which maintains the homeostasis of osmotic pressure in these tissues and integrates different stimuli that translate into $Ca^{2+}$ signals regulating the functions of the respiratory system (Garcia-Elias et al, 2014). In the respiratory airway, the overactivation of TRP channels induces airway constriction, inflammation, sneezing, cough, and mucus secretion, contributing to a defense mechanism of the respiratory system (Parker et al, 1998; Wallace, 2017; Xia et al, 2018). It has been shown that agonists of TRPV4 and hypoosmotic stress result in depolarization of the vagal nerves in humans, mice, and guinea pigs. Conversely, the use of TRPV4 antagonists results in a decrease in cough (Bonvini et al, 2016), which has led to the proposal that TRPV4 is a molecular effector of airway protection.

Roles of TRPV4 in acute lung injury and in acute respiratory distress syndrome have been also suggested. In these processes, there is an increase in the permeability of pulmonary vascular and endothelial barriers (Catravas et al, 2010) likely involving TRPV4 overactivation, because pharmacological inhibition or genetic deletion of this channel prevents lung damage in animal models of ventilator-induced pulmonary injury by liquids (Hamanaka et al, 2007; Bihari et al, 2017). Moreover, experiments where mice were exposed to chemical compounds (i.e., hydrochloric acid or chloride vapors) (Balakrishna et al, 2014) showed that TRPV4 plays a crucial role in injury to the lungs. In the respiratory system, overactivation of TRPV4 affects excitability of bronchopulmonary sensory neurons (Gu et al, 2016) and TRPV4-deficient mice are protected from airway remodeling that occurs in both large and small airways relevant to miscellaneous respiratory diseases including asthma, a chronic inflammatory disease of the upper airways (Gombedza et al, 2017).

This channel has also been linked to the response to contaminating particles, such as those from diesel exhaust (Li et al, 2011), and it has recently been proposed as a pharmacological target for patients with COVID-19, where inhibition of TRPV4 may reduce lethality by contributing to alveolo-capillary barrier preservation (Goyal et al, 2019).

$\beta$-Adrenergic receptors ($\beta$-ARs) are $G_s$ protein–coupled receptors that elicit smooth muscle relaxation and bronchodilation, through activation of a cAMP-dependent signaling pathway (Barisione et al, 2010). In asthma, agonism of $\beta$-ARs by compounds such as salbutamol results in bronchodilation, allowing for better respiratory function.

Although clenbuterol, like salbutamol, is considered a specific $\beta$-AR agonist, it has been shown to inhibit sodium channels in rat skeletal muscle fibers, where $\beta$-ARs are also expressed (Desaphy et al, 2003). Because TRPV4's activation influences the function of respiratory airways, here we studied whether salbutamol and other agonists and antagonists of $\beta$-ARs could regulate the function of this ion channel in vitro. Our results indicate that not only agonists of $\beta$-ARs salbutamol, terbutaline, isoprenaline, metaproterenol, levalbuterol, and clenbuterol inhibit the activation of TRPV4, but also antagonists of these receptors exert inhibitory effects on the channel. Our site-directed mutagenesis data, showing that

salbutamol binds to residues located at an extracellular site on the pore turret of TRPV4, suggest that this inhibitory effect is mediated by direct interaction with this tetrameric ion channel.

## Results

### Salbutamol inhibits activation of TRPV4, but not of TRPV1

The short-acting bronchodilator (SABD) salbutamol, composed of a racemic mixture of R- and S-salbutamol, which is a sympathomimetic amine that functions as an agonist of $\beta$2-AR, was used to perform experiments in a heterologous expression system of TRPV4. In HEK293 cells transfected with human TRPV4 (hTRPV4), we found that application of 300 nM GSK to outside-out membrane patches resulted in the activation of currents (Fig 1A–D, black traces). Experiments where $Ca^{2+}$ was added to the external solution showed that after exposure to salbutamol, 64.5% ± 12.3% (Fig 1E, +120 mV, solid circles) of the current magnitude was inhibited, and hence, only a fraction of GSK-activated current remained after re-exposing the patches to the agonist (Fig 1A and E). Because $Ca^{2+}$ has been shown to exhibit complex modulation properties (potentiation and then inhibition) on TRPV4's activity (Nilius et al, 2004), we performed all electrophysiological experiments, except those shown in Fig 1A and E, in the absence of this ion. When membrane patches were exposed to vehicle only (Tris), there was only a 21.2% ± 9.7% decrease (Fig 1E, +120 mV, empty triangles) in current, as compared to the initial current obtained in the presence of GSK, which could be due to the removal of an intracellular factor required for sustained activity upon membrane patch excision (Garcia-Elias et al, 2013).

To determine whether salbutamol inhibited closed or open TRPV4 channels equally, we measured currents initially activated by 300 nM GSK and then incubated the membrane patches with salbutamol (500 μM, 5 min) either in the absence (Fig 1B) or in the presence (Fig 1C) of the agonist, and then, we reapplied GSK to determine inhibition by salbutamol (purple traces at –120 and +120 mV). We observed that when salbutamol was applied in the absence of GSK, 54.3% ± 14.5% (closed state, Fig 1E, solid triangles, +120 mV) of TRPV4's currents were inhibited, as compared to the 32.5% ± 18.6% inhibition when salbutamol was applied in the presence of the agonist (open state, Fig 1E, solid ties, +120 mV), suggesting that there is state-dependent inhibition of TRPV4-mediated current by salbutamol. In contrast, when rundown in the presence of vehicle and GSK101 was assessed, we found that 16.16% ± 8.3% of the current was decreased (Fig 1E, empty ties). Furthermore, we performed experiments using a related TRP channel, TRPV1, to determine whether salbutamol inhibited this channel also. As shown in Fig 1D and E, capsaicin (250 nM) activated the TRPV1 channel, but the addition of salbutamol (500 μM, orange traces at –120 and +120 mV) did not produce a significant current decrease (2.6% ± 16.8%, Fig 1E, orange squares, +120 mV).

In another set of experiments, we found that GSK-activated currents were also inhibited by 500 μM salbutamol when it was applied intracellularly (Fig S1A and B). Specifically, in inside-out patches, 500 μM salbutamol inhibited 36% ± 22.4% of the currents, as compared to 12.78% ± 12% inhibition with intracellularly applied

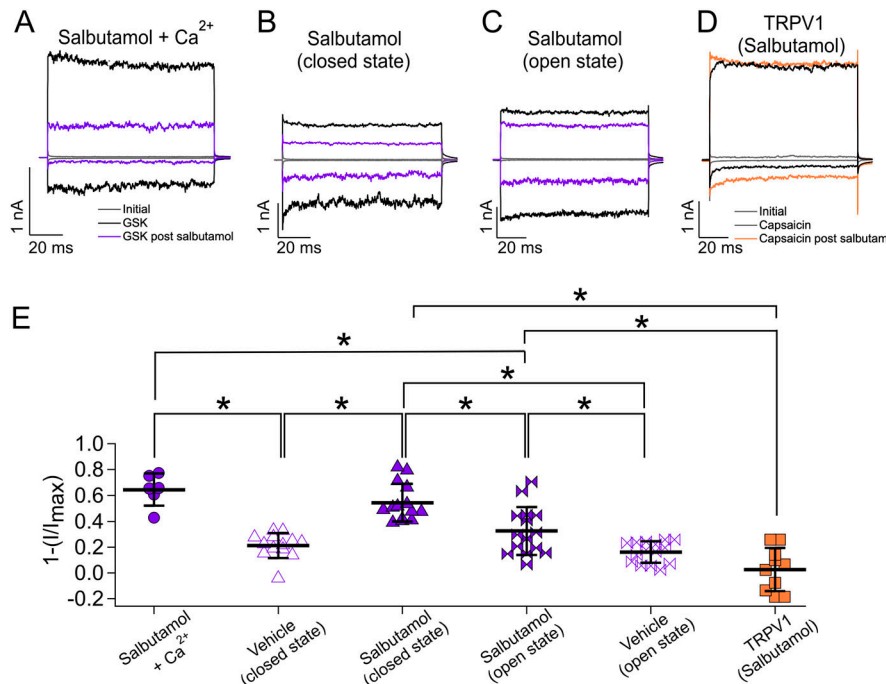

**Figure 1.  Effects of salbutamol on TRPV4 and TRPV1 channels.**

**(A, B, C, D)** Representative traces of currents at +120 and −120 mV for 100 ms from a holding potential of 0 mV in the outside-out configuration. Patches were first exposed to recording solution (gray, leak or initial currents), then to agonists (black traces) GSK (300 nM) for TRPV4 channels or capsaicin (200 nM) for TRPV1, washed with recording solution, and then exposed to 500 μM salbutamol. Current inhibition was measured after exposing the patches to the agonists again (purple traces for TRPV4 and orange for TRPV1). **(E)** Average data for experiments in (A, B, C, D). Data were normalized to the initial value with GSK or capsaicin. *The one-way analysis of variance, followed by Tukey's post hoc, was used for group comparison. Significant differences between means were considered to exist when the $P$-value was less than 0.01. After salbutamol treatment in the presence of $Ca^{2+}$, 64.5% ± 12.3% (n = 6) of currents were inhibited. The percentages of inhibited currents when salbutamol was applied in the closed (absence of agonist) or in the open (presence of agonist) states were 54.3% ± 14.5% (n = 14) and 32.5% ± 18.6% (n = 15), respectively. Application of vehicle only in the closed state produced 21.2% ± 9.7% (empty triangles, n = 13) of the inhibition and 16.16% ± 8.2% (empty ties, n = 15) when vehicle was applied in the presence of GSK101. In the case of TRPV1, 2.6 ± 16.8 (n = 10) of currents were inhibited after treatment with salbutamol (orange squares). *$P$ < 0.0001 for salbutamol + $Ca^{2+}$ versus vehicle; *$P$ < 0.001 for salbutamol + $Ca^{2+}$ versus salbutamol + GSK (open); *$P$ < 0.0001 for vehicle versus salbutamol (closed); *$P$ = 0.0005 for salbutamol (closed) versus salbutamol (open); *$P$ < 0.0001 for salbutamol (closed) versus salbutamol in TRPV1; and *$P$ = 0.0002 for salbutamol (open) versus salbutamol in TRPV1, as indicated by brackets. One-way analysis of variance with Tukey's post hoc test was performed. Source data are available for this figure.

vehicle (Fig S1C, +120 mV, solid and empty circles, respectively). Inhibition by salbutamol applied to the extracellular side of the channel was statistically different (54.3% ± 14.5%, Fig S1C, +120 mV, triangles), as compared to when it was applied from the intracellular side (36% ± 22.4%, Fig S1C, +120 mV, solid circles).

We further examined inhibition of TRPV4 after a 5-min incubation with different concentrations of extracellular salbutamol and found that it is concentration-dependent. The inhibition was quantified by fitting to the Hill equation with an apparent dissociation constant ($K_D$) of 81.3 μM and a Hill coefficient of 1.4 (Fig 2A). It is interesting to note that an inhaler, which can be used three or four times a day, typically contains between 100 and 150 μM/kg salbutamol per dose. Interestingly, nebulized salbutamol has been found to be more effective, as compared to its systemic administration. A possible explanation for this is that there appears to be no direct biotransformation of salbutamol in the lungs (with a half-life between 2 and 7 h), when applied with an inhaler (Gad, 2014). Finally, we performed experiments to determine the time course of TRPV4 inhibition by 500 μM salbutamol at different time points (as detailed in the Materials and Methods section) and found that the current decay could be fitted to a single exponential with a time constant of 225.6 s (Fig 2B). Current decay stabilized after 5 min of exposure of the membrane patches to salbutamol, and it was not measured after 6 min of exposure because current decay, in the presence of only vehicle at 7 min, was already 49.96% ± 25.97% (n = 7). Finally, we also measured recovery from inhibition by measuring initial currents with GSK101 (300 nM), then washing GSK101 off to apply salbutamol (500 μM) for 5 min and reactivating the channels by reapplying GSK101 for up to 10 min. The data in Fig 2C show that

right after treatment with salbutamol 500 μM (0–60 s), some current was activated, and then, it was not further recovered in the presence of GSK101, possibly because of a combination of this inhibition being irreversible and rundown of the activity of the channel with excision of the membrane patch.

## Effects of short- and long-acting bronchodilators on TRPV4 activation

It has been shown that the minimal required chemical structure for activation of β-ARs is an aromatic ring system and an aliphatic amino group (Kolb et al, 2009). Structural components such as the chiral β-OH group present in salbutamol and in endogenous agonists (i.e., epinephrine and norepinephrine) (Swaminath et al, 2005) also allow for better binding to β-ARs.

Different agonists are thought to either partially stabilize or fully activate β-ARs through interactions of different chemical groups with different residues in the receptors (Yao et al, 2006). In the case of salbutamol, these functional groups are the OH groups present in the aromatic ring, which interact with residues S203, S204, S207, D113, and N312 in the β2-AR (Strader et al, 1989; Johnson, 1998). Other SABDs, with structural similarities to salbutamol (Fig 3A–E), include levalbuterol (which is composed only of R-salbutamol), terbutaline, isoprenaline, and metaproterenol. To ascertain whether these compounds could also inhibit TRPV4, we performed experiments in which we first activated TRPV4-mediated currents in the presence of GSK (300 nM), then applied each different SABD, and measured the inhibited current after the second application of GSK. We used a concentration of 500 μM for each compound, with which

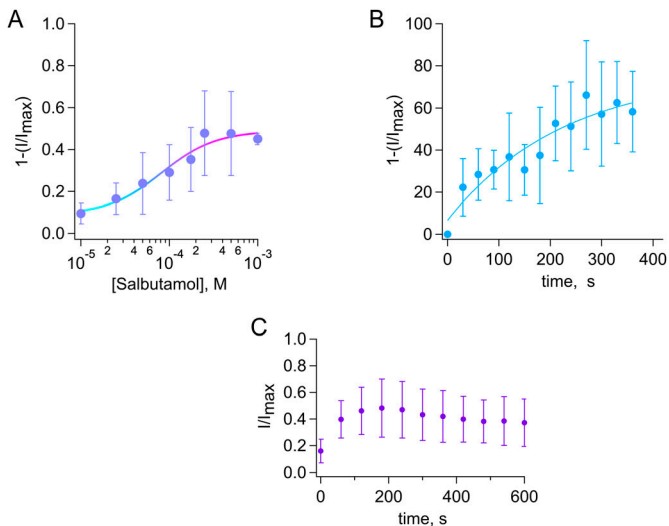

**Figure 2.   Dose-dependent inhibition of TRPV4 channels by salbutamol.**
**(A)** Dose–response for inhibition by salbutamol (5 min) of currents activated by GSK 300 nM (+120 mv). Smooth curve is a fit with the Hill equation ($K_D$ = 81.3 μM and Hill coefficient = 1.4). A single salbutamol concentration in the absence of GSK was tested per outside-out membrane patch, and the remaining GSK-activated current was normalized to the current obtained with GSK initially (n = 3–12 for each concentration point). **(B)** Time course of inhibition by 500 μM salbutamol, $\tau$ = 225.6 s (n = 8–10 for each time point). Data were obtained at –60 mV and fit to a single exponential. **(C)** Recovery from inhibition by salbutamol (500 μM). Current inhibition is irreversible, remaining fraction of currents are shown, and the zero time point represents currents in the presence of GSK101 right after application of salbutamol 500 μM for 5 min.
Source data are available for this figure.

we achieved maximal inhibition with salbutamol, and like salbutamol 500 μM (Fig 3A and F, 52.12% ± 16.3% current inhibition after salbutamol), the other compounds also produced similar TRPV4 inhibition, evidenced after the second application of GSK. Levalbuterol (Fig 3B and F) resulted in a 42.6% ± 19.4% inhibition of TRPV4 currents, whereas terbutaline (Fig 3C and F), isoprenaline (Fig 3D and F), and metaproterenol (Fig 3E and F) resulted in a 54.4% ± 15%, 51.6% ± 25.9%, and 51.7% ± 28% current inhibition, respectively, at a concentration of 500 μM. Taken together, these data show that several agonists of β-ARs can inhibit to similar levels the activity of the TRPV4 channel.

As mentioned before, all the above compounds are considered SABDs, and their chemical structures reveal that they possess certain features such as an aromatic ring, where hydroxymethyl and hydroxyl groups are present in distinct positions (Swaminath et al, 2005).

Clenbuterol is a long-acting β-AR agonist, and structurally, it varies from salbutamol in that it does not possess the hydroxymethyl group and hydroxyl group at the third and fourth positions of the benzene ring, respectively, but contains chlorine atoms at the third and fifth positions and an amine group at the fourth position of the benzene ring. To determine whether clenbuterol also inhibits TRPV4, we performed electrophysiological experiments where we first activated TRPV4 with GSK (300 nM, black traces), then applied clenbuterol (500 μM, blue traces) for 5 min, and, finally, we measured the inhibited current by reapplying GSK. The results in Fig 4A, B, and E show that clenbuterol (blue traces in Fig 4B) produced a more pronounced inhibition of TRPV4, as compared to salbutamol (purple traces in Fig 4A), because current inhibition after treatment was of 52.12% ± 16.3% for salbutamol versus 71.7% ± 11.2% for

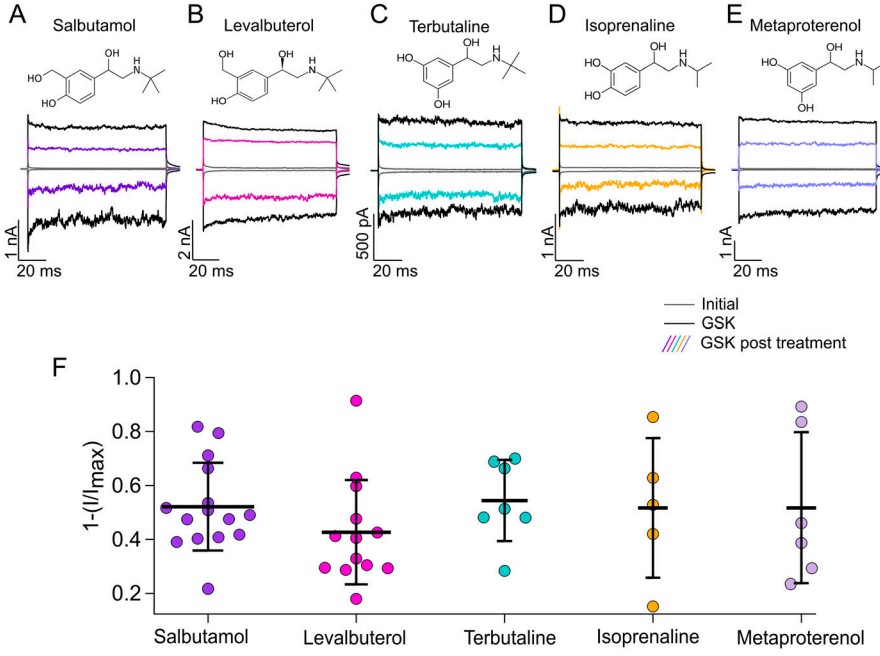

**Figure 3.   Effects of short-acting bronchodilator on TRPV4 currents.**
**(A, B, C, D, E)** Representative traces of currents at +120 and –120 mV, obtained as in Fig 1A for different compounds. Trace colors represent gray for leak or initial currents, black for GSK 300 nM, and different colors for GSK after exposure of outside-out membrane patches to 500 μM of salbutamol, levalbuterol, terbutaline, isoprenaline, or metaproterenol.
**(F)** Average data for experiments in (A, B, C, D, E). Data were normalized to the initial value with GSK. The percentages of inhibited currents after different treatments are as follows: 52.12% ± 16.3% after salbutamol (n = 15), 42.6% ± 19.4% after levalbuterol (n = 13), 54.4% ± 15% after terbutaline (n = 7), 51.6% ± 26% after isoprenaline (n = 5), and 51.7% ± 28% after metaproterenol treatment (n = 6). No statistically significant differences were found with one-way analysis of variance.
Source data are available for this figure.

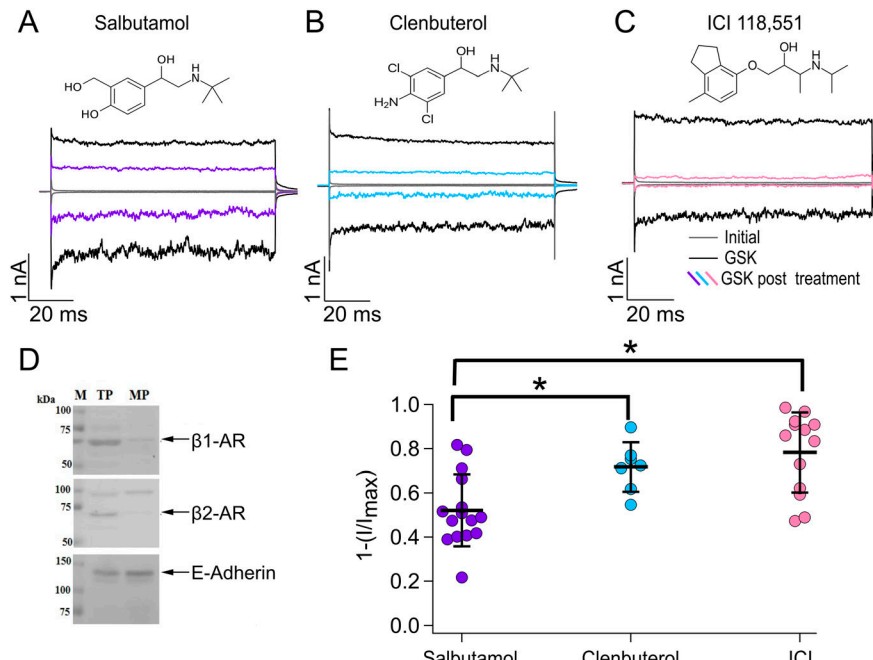

**Figure 4. Agonists and antagonists of β-adrenergic receptors.**
**(A, B, C)** Representative traces of currents at +120 and −120 mV, obtained as in Fig 1A for different compounds. Trace colors represent gray for leak or initial currents, black for GSK 300 nM, and different colors for GSK after exposure of outside-out membrane patches to 500 μM of salbutamol, clenbuterol, or ICI. **(D)** Total and membrane proteins were analyzed by Western blot for immunodetection of β1- and β2-adrenergic receptors; E-cadherin was used as a positive control of plasma membrane protein. **(E)** Average data for experiments in (A, B, C). Data were normalized to the initial value with GSK. The percentages of inhibited currents after different treatments are as follows: 52.12% ± 16.3% after salbutamol (n = 15), 71.7% ± 11.2% after clenbuterol (n = 7), and 78.3% ± 18% after ICI (n = 13). *P = 0.0328 for salbutamol versus clenbuterol and *P = 0.0005 for salbutamol versus ICI, as indicated by brackets. One-way analysis of variance with Tukey's post hoc test was performed.
Source data are available for this figure.

clenbuterol (Fig 4E). These data indicate that, like what happens in β-ARs, differences in the chemical structures of agonists of these receptors also yield differences in their effects on TRPV4 activity.

### Antagonism of β-adrenoreceptors also results in TRPV4 inhibition

To gain insight into the mechanism by which bronchodilators can inhibit TRPV4's activity, we tested an antagonist of β2-ARs, ICI-118,551 (ICI), which has been described as a potent and selective inhibitor of β2-ARs (Bilski et al, 1983) and for which, to the best of our knowledge, no therapeutic use in humans has been yet assigned. Using our heterologous TRPV4 expression system of HEK293 cells, which endogenously expresses β1- and β2-ARs (Fig 4D), we performed electrophysiology experiments where we tested the hypothesis of whether TRPV4 inhibition was due to the activation of these receptors. In this scenario, we would expect the antagonist not to produce inhibition of the TRPV4 ion channel in response to GSK, as we observed with agonists of β-ARs. Notably, as shown in Fig 4C (pink traces) and Fig 4E, we found that ICI produced even more inhibition of TRPV4's activation in response to GSK (black traces) than that of salbutamol because 78.3% ± 18% of the current was inhibited after ICI versus 52.12% ± 16.3% with salbutamol.

Because ICI was effective in inhibiting TRPV4 activation by GSK, we further characterized its effects on TRPV4 by performing experiments to evaluate inhibition when ICI was applied to open channels (in the presence of GSK) or when it was applied to closed channels (in the absence of GSK). ICI inhibited TRPV4 (pink traces) very similarly whether it was applied in the presence or absence of the agonist, because 81.7% ± 17.6% and 77.3% ± 18.5% of the current were inhibited, respectively (Fig 5A and B). Moreover, as evidenced from the dose–response in Fig 5C, ICI produced more inhibition of

the channel, as compared to salbutamol, with a $K_D$ of inhibition of the channel of 3.9 μM.

Together, these data suggest that down-regulation of TRPV4 activity by agonists of β-ARs is independent of these receptors. To further substantiate this conclusion, we performed experiments where we incubated TRPV4-expressing HEK293 cells for 24 h with 11 μM ICI and measured GSK-induced TRPV4 currents from excised HEK293 cell membrane patches in the presence of a concentration near the $K_D$ of salbutamol (100 μM), which is expected to inhibit around 25% of the total current because the maximal concentration of 500 μM salbutamol inhibits 50% of current. Fig 5D and E shows that antagonism of β2-ARs did not affect inhibition of TRPV4 by salbutamol (23.6% ± 17.3% with salbutamol versus 23.4% ± 18.96% with ICI preincubation). These results are in accordance with what we would expect if salbutamol was acting directly on TRPV4 and independently of β2-AR–associated signaling pathways. These results prompted us to assess whether, in fact, salbutamol and the other tested compounds could be directly interacting with TRPV4.

### Salbutamol stabilizes the closed state of TRPV4

We addressed the mechanism of salbutamol inhibition of TRPV4 channels using outside-out membrane patches. Single-channel recordings showed that upon application of 300 nM TRPV4 agonist GSK, the channel stays open (O) (Fig 6A and B, blue traces and symbols). However, co-application of salbutamol and GSK resulted in a decrease in the open probability with the channel transitioning to the closed state (C) more frequently (Fig 6A and B, purple traces and symbols). Single-channel recordings also confirmed that salbutamol did not alter the single-channel current, as the amplitude remains similar before (Fig 6C) and after application of salbutamol (Fig 6D). All the above observations are consistent with the interpretation that

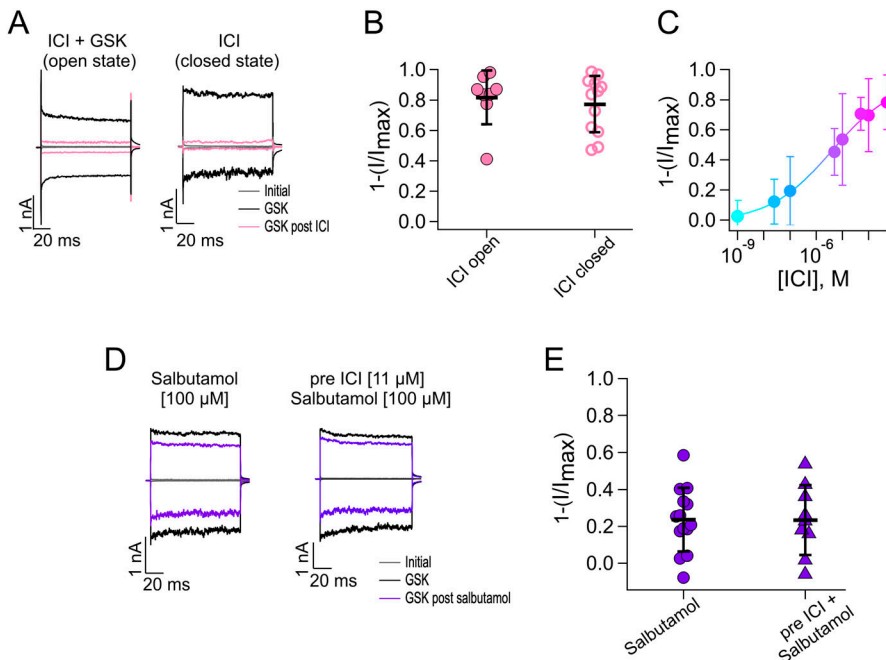

**Figure 5.  Salbutamol inhibits TRPV4 independently of β-adrenergic receptors.**
**(A)** Representative traces of currents at +120 and −120 mV. Gray traces are leak or initial currents, and black traces are with GSK 300 nM before treatment with ICI and after ICI (500 μM) applied in the presence of GSK (open state, pink traces) and absence of GSK (closed state, pink traces). **(B)** Average data for experiments in (A). Data were normalized to the initial value with GSK. The percentages of inhibited currents after different treatments are as follows: 81.7% ± 17.6% after ICI + GSK (n = 8), and 77.3% ± 18.5% after ICI only (n = 12). **(C)** Dose–response for inhibition by ICI (5 min) of currents activated by GSK 300 nM (+120 mv). Smooth curve is a fit with the Hill equation (K$_D$ = 3.9 μM and Hill coefficient = 0.36). A single ICI concentration was tested per outside-out membrane patch, and the remaining GSK-activated current was normalized to the current obtained with GSK initially (n = 6–13 for each concentration point). **(D)** Representative traces of experiments where patches were exposed to salbutamol or salbutamol in cells previously incubated with ICI for 24 h. Both compounds were used at concentrations near their K$_D$ values. **(E)** Average data for experiments in (D). Data were normalized to the initial value with GSK. The percentages of inhibited currents after different treatments are as follows: 23.6% ± 17.3% after salbutamol (n = 14) and 23.4% ± 18.96% after preincubation with ICI (n = 9). **(B, E)** Data
were not statistically significant with an unpaired *t* test for data in (B, E). Source data are available for this figure.

salbutamol allosterically stabilizes TRPV4 channel in the closed state or produces a very slow open-channel block.

## Interaction sites of salbutamol with TRPV4

In β2-ARs, R-salbutamol binds in the catecholamine pocket, with its secondary and β-hydroxyl groups forming hydrogen bonds with D, N, and S residues and/or van der Waals contacts with F, W, T, S, and V residues (Katritch et al, 2009).

Molecular docking simulations were performed to gain insight into the structural mechanism that governs the association between TRPV4 and salbutamol. Exploring the entire channel surface, we identified most of the salbutamol (R-isomer) conformations interacting with the extracellular region of TRPV4. The search conformational space was then delimited to this area, finding four clusters, totalizing 1,000 TRPV4–salbutamol-docked conformations (Fig 7A). The lowest scoring (lowest energy) conformation of salbutamol in clusters 1, 2, 3, and 4 interacted with residues E514-S563-Y567, S667-V693-I696-I697, S687-S688-D682, and A489-P493-S634-N637, respectively (Fig 7B). Serine residues were demonstrated to be particularly relevant to coordinate the hydroxyl groups in the salbutamol molecule.

To verify the functional relevance of TRPV4 residues that could potentially interact with salbutamol, based on what is known for β-ARs and the data we obtained from the in silico experiments, we performed electrophysiological experiments to assess inhibition of TRPV4 by salbutamol in the following mutant channels: S557A and S563A in S3, E572A in the S3-S4 linker, S630A and S634A in S5, and S667A, D682A, S687A, S688A, and T689A in the pore, as compared to WT channels.

As shown in Fig S2, TRPV4 mutant channels with single-residue substitutions: S557A, S563A, E572A, S630A, S634A, D682A, and T689A, produced proteins that responded similar to WT TRPV4 after addition

of 500 μM salbutamol for 5 min and reactivation of currents with 300 nM GSK (different color traces). Hence, we concluded that these sites were not responsible for interactions of salbutamol with TRPV4. We also tested the mutation S688A, which was identified by our MD simulations as a possible candidate for interaction of salbutamol with TRPV4. However, this mutation gave rise to channels with enhanced rundown that could not be recovered with GSK101 after 5 min in the presence of recording solution only (Fig S3); consequently, effects of salbutamol on this mutant could not be evaluated.

However, substitution of residues S667A and S687A, located extracellularly at the entrance of the pore, produced ion channels that were less inhibited by salbutamol (Fig 8A and B), as compared to the WT TRPV4 channels. The TRPV4-S667A mutant channels exhibited 43.2% ± 12.9% inhibition (squares, Fig 8B) of currents activated with GSK after treatment with salbutamol, whereas TRPV4-S687A channels exhibited a 31% ± 14.5% inhibition of currents activated with 300 nM GSK (triangles, Fig 8B) after salbutamol, as compared to WT channels (60.1% ± 19.6%; circles, Fig 8B). Finally, mutant channels containing mutations of S667A and S687A together were tested for inhibition with salbutamol and the results show that the double mutant only exhibits 10.9% ± 13.4% inhibition (bow ties, Fig 8B). WT TRPV4 channels and S667A, S687A, and S667A-S687A mutants all activated in a dose-dependent fashion in response to GSK (Fig 8C). These data indicate that serine residues at positions 667 and 687 are important for salbutamol interaction with TRPV4.

## Discussion

The activity of the TRPV4 ion channel has been widely linked to several physiological and pathophysiological processes. This ion

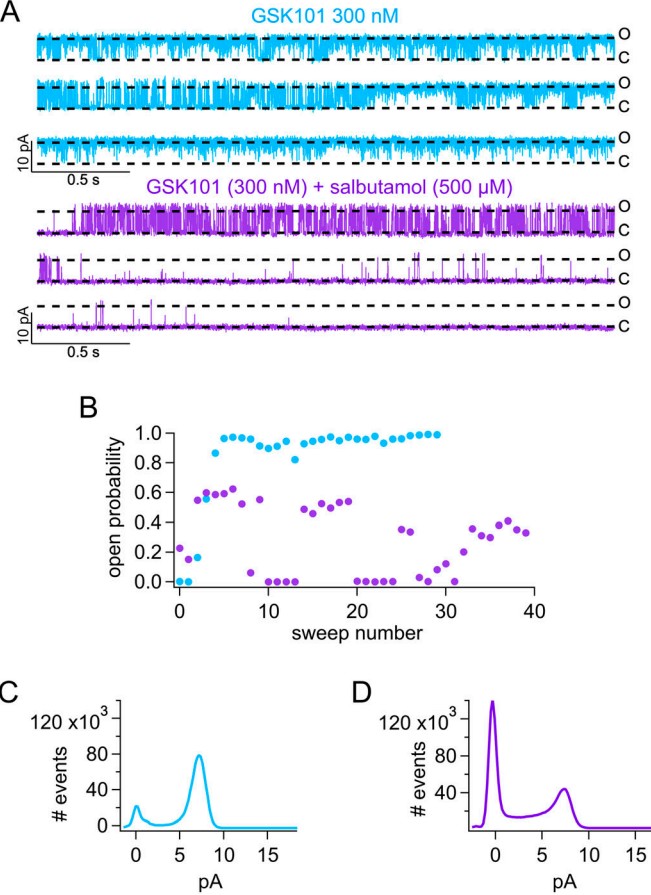

**Figure 6. Salbutamol produces a decrease in the open probability of TRPV4.**
**(A)** Representative single-channel recordings in outside-out membrane patches show that in the presence of GSK (300 nM), TRPV4 is mostly in the open (O) state (blue traces). When salbutamol (500 μM) is added in the presence of 300 nM GSK, the channel starts to close (C), leading to a decrease in the open probability ($P_o$, purple traces). **(B)** $P_o$ of TRPV4 calculated for the representative trace in (A). Blue symbols correspond to the $P_o$ calculated for each sweep (+60 mV) in the presence of only GSK, and the purple symbols are for sweeps in the co-application of GSK + salbutamol. **(C, D)** Co-application of salbutamol decreases $P_o$ without significantly changing the unitary channel current, as shown for GSK alone ((C), blue) and GSK + salbutamol ((D), purple).

channel is expressed in various organs and contributes to their function (Nilius & Owsianik, 2010; White et al, 2016; Rosenbaum et al, 2020). Specifically, of interest to the present study is that over-expression or increased activity of TRPV4 leads to changes in the alveolo-capillary barrier (Gombedza et al, 2017). Moreover, it has been suggested that TRPV4 plays a role in the remodeling and obstruction of airways through influencing the proliferation of cells in the lungs (Zhao et al, 2014) and, hence, contributing to the presence of chronic asthmatic conditions. Activation of TRPV4 has been shown to result in increased entrance of $Ca^{2+}$, leading to the proliferation of smooth muscle cells and influencing chronic asthmatic conditions (Zhao et al, 2014). In addition, inhibition of the channel with GSK3491943 and GSK3527497 can mitigate TRPV4-induced pulmonary edema (Cheung et al, 2017; Brnardic et al, 2018). Thus, it stems that inhibition of this channel in the airways (Balakrishna et al, 2014), as has also been proposed for

patients with SARS-CoV-2, which are subject to lung barrier damage because of mechanical overstimulation by respirators, may be beneficial under certain scenarios (Kuebler et al, 2020).

Inhibitors of TRPV4's activity have been described and tested in animals. Also, assays in humans have determined their safe use (Mizuno et al, 2003; Everaerts et al, 2010; Vincent & Duncton, 2011; Thorneloe et al, 2012; Kanju et al, 2016; Yin et al, 2016; Cheung et al, 2017; Pero et al, 2018; Brooks et al, 2019; Goyal et al, 2019; Achanta & Jordt, 2020; Kuebler et al, 2020; Yang et al, 2022). Here, we report for the first time that commonly used compounds to treat broncho-spasms inhibit TRPV4 currents. Although it is established that these compounds interact with β-ARs (Nelson, 1995), our data show that TRPV4 channels respond to salbutamol and other related chemicals, albeit at higher concentrations than those to which β-ARs are sensitive.

Salbutamol is a chiral compound usually found as a 50:50 racemic mixture of two enantiomers (stereoisomers), which are molecules with non-superimposable mirror images. It has one chiral center in the ethanolamine tail, resulting in the two enantiomers (S)- and (R)- for the hydroxyl-group position. Our experiments using the racemic mixture of salbutamol show partial inhibition of TRPV4 (Figs 1 and 2) and the R-salbutamol enantiomer (levalbuterol; Fig 3) produces a similar inhibitory effect to that observed with the racemic mixture. We do not know whether the S-enantiomer inhibits TRPV4, but it is noteworthy that S-salbutamol alone is not clinically used because it has been reported either to not have any kind of effect on asthmatic patients or to produce bronchoconstriction (Templeton et al, 1998; Gumbhir-Shah et al, 1999). It is also important to mention that most of the experiments in this study were performed in the absence of $Ca^{2+}$ because, as stated above, this ion can produce inhibition of TRPV4. In this sense, it is hard to hypothesize how the compounds tested here would affect the function of TRPV4 under physiological conditions because several scenarios are viable with the channel possibly exhibiting smaller currents because of the presence of $Ca^{2+}$, but also being subject to potentiation of its activity by changes in its phosphorylation state (Wegierski et al, 2009), changes in PIP2 concentration (Garcia-Elias et al, 2013; Harraz et al, 2018), and other changes in the cellular environment.

One previous study suggested that clenbuterol but not salbutamol inhibits skeletal muscle sodium (hSkM1) channels and that such inhibition is independent of β-ARs (Desaphy et al, 2003). To our knowledge, no other ion channel has been shown to be directly modulated by agonists of β-ARs, but an antagonist (ICI) of these receptors that is not for human use has been shown to inhibit the human ether-a-go-go-related gene potassium ion channels, through a mechanism that seems to involve pore block (Dupuis et al, 2005).

Our data show that TRPV4, but not the related TRPV1 channel, is partially inhibited by salbutamol in a dose- and time-dependent fashion, with some preference for the extracellular region of the channel. We demonstrate that other SABDs, which display a higher affinity for β2-ARs (Barisione et al, 2010), such as levalbuterol, terbutaline, isoprenaline, and metaproterenol, and a long-acting β-AR agonist, clenbuterol, are inhibitors of TRPV4. This suggests that chemical structural features shared by these compounds allow for TRPV4 inhibition. Importantly, antagonism of β2-ARs with ICI

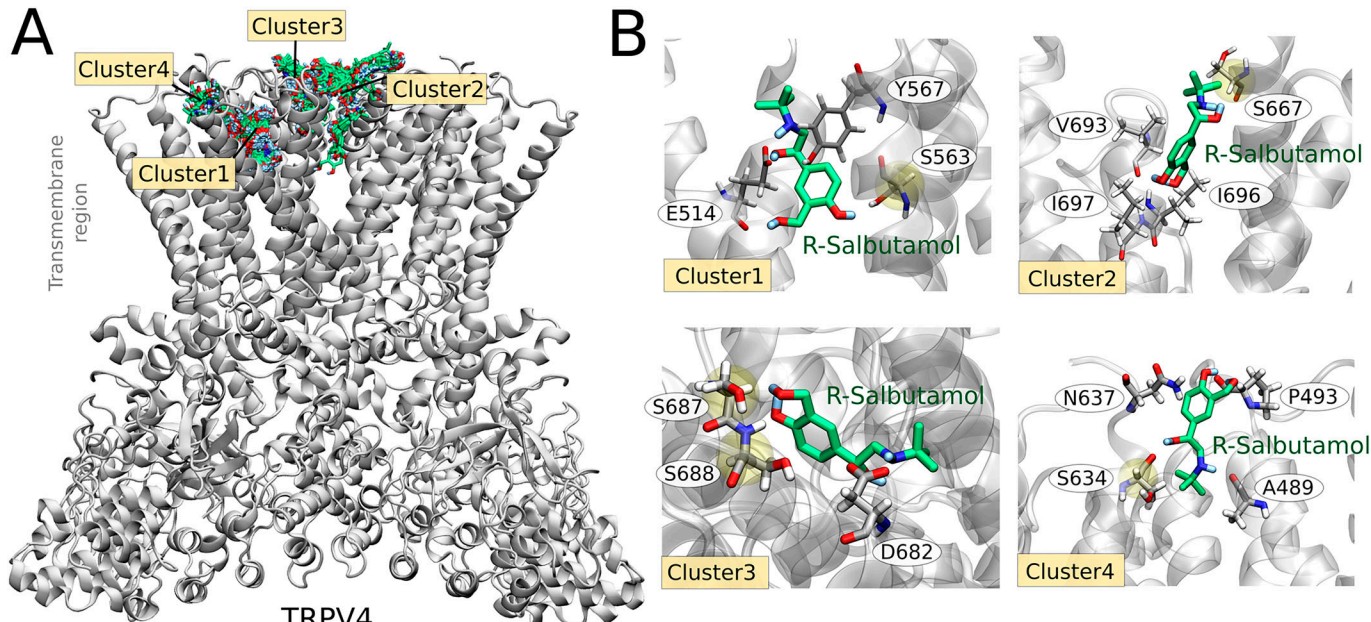

**Figure 7.   Computational modeling of TRPV4–salbutamol association.**
**(A)** Binding conformations of salbutamol (R-isomer), derived from docking simulations, into the human TRPV4 channel (gray cartoon representation). The salbutamol molecule is represented as sticks colored by atom types—C in green, O in red, N in blue, and H in white. Four clusters of docked conformations were obtained. **(B)** Lowest score docking pose in each cluster. TRPV4 residues at 5 Å of the salbutamol molecule are shown as sticks (C in gray).

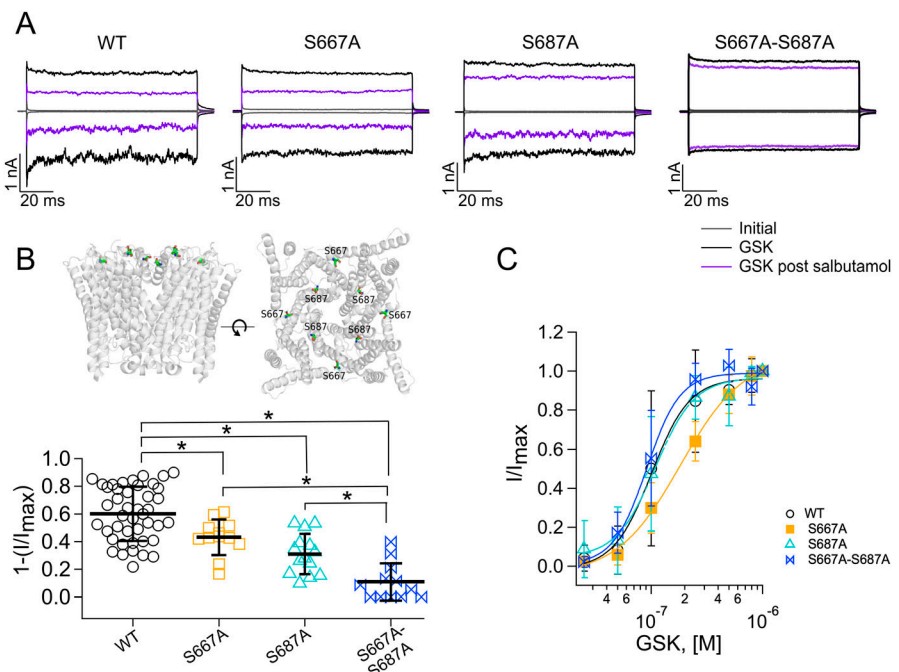

**Figure 8.   Two residues in the pore region mediate TRPV4 inhibition by salbutamol.**
**(A)** Representative traces of currents at +120 and −120 mV, obtained as in Fig 1A. Trace colors represent gray for leak or initial currents, black for GSK 1 μM, and purple for GSK after exposure of outside-out membrane patches to 500 μM of salbutamol for WT and mutant TRPV4 channels. **(B)** After salbutamol treatment, WT TRPV4 (n = 43) channels displayed a 60.1% ± 19.6% inhibition of currents, whereas 43.2% ± 12.9%, 31% ± 14.5%, and 10.9% ± 13.4% inhibition was observed for the S667A (n = 12), S687A (n = 14), and S667A-S687A (n = 12) mutants, respectively. Mutated residues are displayed as spheres in each subunit in a side and top view. Only the transmembrane region of the human TRPV4 ion channel (PDB code: 7AA5) is exhibited. *P = 0.0178 for WT TRPV4 versus TRPV4-S667A; *P < 0.0001 for WT TRPV4 versus TRPV4-S687A; *P < 0.0001 for WT TRPV4 versus TRPV4-S667A-S687A; *P < 0.0001 for TRPV4-S667A versus TRPV4-S667A-S687A; and *P < 0.02 for TRPV4-S687A versus TRPV4-S667A-S687A, as indicated by brackets. One-way analysis of variance with Tukey's post hoc test was performed. **(C)** Dose–response for activation of WT, S667A, and S687A TRPV4 channels by GSK (+120 mv). Smooth curve is a fit with the Hill equation. The fits yield $K_D$ = 101 nM and Hill coefficient = 2.5 for WT channels, $K_D$ = 182 nM and Hill coefficient = 1.5 for the S667A mutant channels, $K_D$ = 108 nM and Hill coefficient = 2.4 for S687A mutant channels, and $K_D$ = 91 nM and Hill coefficient = 2.9 for S667A-S687A mutant channels. A GSK concentration was tested per outside-out membrane patch (n = 4–11 for each concentration point). Source data are available for this figure.

does not preclude inhibition of TRPV4 by salbutamol, supporting a direct action of this compound on TRPV4. In fact, inhibition of TRPV4 by ICI is more efficient, as compared to salbutamol, and TRPV4 exhibits inhibition of the channel at lower concentrations than those shown to affect human ether-a-go-go-related gene channels (Dupuis et al, 2005).

These observations led us to screen possible interaction sites for salbutamol with TRPV4. Computational modeling and

electrophysiological experiments revealed the importance of residues S667 and S687 for inhibition of TRPV4 by salbutamol. Moreover, the mutation S688A renders the channel more sensitive to rundown, as demonstrated by our data where GSK-activated currents could not be recovered after 5 min in the presence of recording solution only (Fig S3). Further studies will be required to determine whether this residue undergoes posttranslational modification that is important for channel function, which is eliminated by the inserted point mutation to alanine, or whether the mutation just renders the channel energetically unstable.

We also identified that residues in the S3, S3-S4 linker, S5, and pore could mediate salbutamol's binding to TRPV4. By testing inhibition of S557A, E572A, S630A, S563A, S634A, and D682A by salbutamol, no significant differences among inhibition of these mutants and the WT TRPV4 channels were found.

It is noteworthy that molecular mechanisms underlying the inhibition of TRPV4 by its antagonists have only been described for some compounds. For example, it has been described that the antagonist HC067047 binds to residues in the S2–S3 linker (D542), in S4 (M583 and Y587), and in S5 (D609 and F613) (Doñate-Macian et al, 2022). By further testing sites identified through our molecular docking experiments, we found for salbutamol that mutation of residues S667 and S687 to alanine produces TRPV4 channels with a diminished response to this compound. These residues are located near the mouth of the pore of the channel; hence, the possibility of salbutamol acting as a pore blocker arises. One would expect a blocker of the pore to produce a decrease in the single-channel amplitude (or conductance). Nonetheless, our single-channel experiments show that the single-channel amplitude is not decreased by exposure of TRPV4 channels to salbutamol, but that the channels transition to the closed states in the presence of the bronchodilator, which is consistent with an allosteric mechanism rather than that of pore block.

80 μM of salbutamol is equivalent to 19 μg/ml (salbutamol molecular weight = 239.311 g/mol). Several studies have reported plasma levels of salbutamol after its intake (inhaled or oral) in the treatment of acute asthma that range between 1.75 and 18.77 ng/ml in adults (Elers et al, 2010) and between 2.77 and 18.22 ng/ml in infants (Rotta et al, 2010), but in lung edema fluid from patients treated with salbutamol, the mean concentration reaches 700 ng/ml (Atabai et al, 2002). Moreover, levels of salbutamol in the urine during asthma treatment were shown to increase up to 2,422.2 ng/ml (Elers et al, 2010). Thus, the micromolar concentrations that we found inhibit TRPV4 suggest that some tissues receive doses of salbutamol at which TRPV4 is inhibited.

In summary, here we show that TRPV4, a relevant ion channel for airway function, is partially inhibited by several widely used bronchodilators, which have been classically associated with their actions through $\beta$-ARs, and that, for inhibition of TRPV4 by salbutamol, the mechanism involves allosteric modulation of the ion channel through interactions by, at least, two residues located in the outer pore region of the protein. The present study provides insight into the molecular mechanisms that down-regulate TRPV4 activity, a physiologically important ion channel. $\beta$-Adrenergic medicines such as salbutamol have been contemplated and studied for treatment of spinal muscular atrophy (SMA), with encouraging results, albeit the mechanism by which they function

remains unknown. Future studies will be required to assess whether salbutamol could be of importance in the treatment of disease in which TRPV4 gain-of-function mutations can be present, such as SMA, Charcot–Marie–Tooth disease, and other pathologies. For example, it has been reported that in children with SMA, which can present defects in respiratory muscle strength, treatment with oral salbutamol benefits respiratory function (Kinali et al, 2002; Pane et al, 2008; Khirani et al, 2017).

## Materials and Methods

### Cell cultures

HEK293 cells (ATCC CRL-1573) were cultured in a complete growth medium containing Dulbecco's modified Eagle's medium with high glucose (DMEM; Gibco) complemented with 10% fetal bovine serum (HyClone) and 100 U/ml of penicillin–streptomycin (Gibco). Cell cultures were maintained in a humidified incubator at 37°C with an atmosphere of 95% of air and 5% of $CO_2$. Cells were subcultured every 3 d using 0.25% (w/V) trypsin–EDTA solution (Gibco).

### Western blot

Total and membrane protein extracts of HEK293 cells were obtained using the Pierce Cell Surface Protein Isolation Kit (89881; Thermo Fisher Scientific). Then, 40 μg of protein (total and membrane) was separated by electrophoresis on a 7% SDS–polyacrylamide gel and transferred to a PVDF membrane.

Blots were preincubated with 5% low-fat milk in TBS–0.1% Tween for 1 h at room temperature and incubated overnight at 4°C with $\beta$1-AR (ab3442) (Sun et al, 2021) or $\beta$2-AR (ab182136) (Cellini et al, 2021) antibodies from Abcam or E-cadherin antibody (3195) (Ye et al, 2022) from Cell Signaling Technology (as a positive control for membrane proteins) diluted 1:1,000 for each case. After washing of the primary antibodies, binding was visualized using a secondary horseradish peroxidase (Thermo Fisher Scientific)–labeled anti-rabbit antibody (for adrenergic receptors; 1:10,000 dilution) and anti-mouse antibody (for E-cadherin; 1:10,000 dilution) and enhanced with diaminobenzidine at 100 μg/ml in TBS with 30% $H_2O_2$.

### Transient cell transfection and patch-clamp experiments

For patch-clamp experiments, the cells were grown on coverslips and transfected with the pEGFP-N3 plasmid together with either the human WT or mutant TRPV4 channels for identification of successfully transfected cells. The jetPEI Polyplus-transfection reagent was used to transfect HEK293 with human TRPV4 channels (500 ng of plasmid for excised patch-clamp or 20 ng for single-channel experiments), per the manufacturer's instructions as previously described (Chen et al, 2021).

TRPV4 currents from transiently transfected HEK293 cells were recorded using the patch-clamp technique in the inside-out and outside-out configurations, as indicated for different experiments (Hamill et al, 1981). Solutions were changed with an RSC-200 rapid solution changer (Molecular Kinetics). The recording solutions

contained the following: 130 mM NaCl, 3 mM Hepes (pH 7.2), and 1 mM EDTA for the bath and pipette, and 130 mM NaCl, 3 mM Hepes, and 2 mM $CaCl^{2+}$ (pH 7.2) when experiments were performed in the presence of $Ca^{2+}$, as indicated. Because the channel is inhibited by $Ca^{2+}$, all experiments (except those shown in Fig 1E) were obtained in the absence of $Ca^{2+}$ to avoid a current decrease (Voets et al, 2002). GSK1016790A (GSK; Sigma-Aldrich) was prepared in DMSO to a 15.25 mM concentration for the stock, which was kept at –20°C.

Salbutamol or albuterol ($\alpha_1$-[(tert-butylamino)methyl]-4-hydroxy-1,3-benzenedimethanol hemisulfate salt), terbutaline (5-[2-[(1,1-dimethylethyl)amino]-1-hydroxyethyl]-1,3-benzenediol hemisulfate salt), levalbuterol ([R]-salbutamol hydrochloride), isoprenaline (1-[3′,4′-dihydroxyphenyl]-2-isopropylaminoethanol hydrochloride), metaproterenol ($\alpha$-[(isopropylamino)methyl]-3,5-dihydroxybenzyl alcohol hemisulfate salt), clenbuterol (4-amino-$\alpha$-[t-butylaminomethyl]-3,5-dichlorobenzyl alcohol hydrochloride), and ICI 118,551 ([2R,3R]-rel-3-isopropylamino-1-[7-methylindan-4-yloxy]-butan-2-ol hydrochloride) were all purchased from Sigma-Aldrich, and stock solutions were prepared in water and then diluted in recording solutions immediately before use in experiments. pH was measured after addition of compounds to ensure that it was not modified in the recording solutions to which membrane patches containing TRPV4 channels were exposed.

Experiments were performed at room temperature (24°C). Mean current values were measured after channel activation had reached the steady state (~2 min). Currents were obtained using voltage protocols, where the holding potential was 0 and 10 mV steps from –120 to 120 mV for 100 ms, as indicated for each experiment in the figure legends. Borosilicate glass was used for pipette fabrication (5 MΩ). Currents were low-pass-filtered at 2 kHz and sampled at 10 kHz with an EPC 10 amplifier (HEKA Elektronik) and were plotted and analyzed with IGOR Pro (WaveMetrics, Inc.).

Initial or leak currents were obtained in the absence of agonist, at a given voltage, and GSK was added for 90 s to activate TRPV4 channels. Then, GSK was washed off the membrane patches and salbutamol (or other chemicals, as indicated) was applied for 5 min either in the presence or in the absence of GSK. Finally, currents were remeasured after 90 s of GSK, to assess inhibition of currents.

Dose–response curves for the inhibition of 300 nM GSK-induced TRPV4 currents were performed by applying a given concentration of salbutamol or ICI in the absence of GSK to outside-out excised membrane patches of HEK293 cells expressing hTRPV4 channels. A single concentration of the compounds was tested in each membrane patch, and the data of several patches at a given concentration were pooled. All currents were measured at a voltage of +120 mV. Data were normalized to the currents initially obtained in the presence of only 1 μM GSK. The Hill equation (Equation (1)) was fitted to the data as previously described (Morales-Lázaro et al, 2016) to estimate the steepness of the curve, n, and the apparent dissociation constant, $K_D$.

$$\frac{I}{I_{max}} = \left( \frac{1}{1 + \frac{[X]}{[K_D]}} \right)^n \qquad (1)$$

Time courses of inhibition by 500 μM salbutamol were obtained using continuous pulses at –60 mV. Each time point was obtained by averaging several patches, first exposed to 300 nM GSK, washed, and then exposed to salbutamol in the absence of GSK, and then, inhibition of currents was measured by exposing the patches to 300 nM GSK again. Results are expressed as the fraction of inhibited currents, which was obtained by dividing the currents obtained at –60 mV after treatment with salbutamol by the initial currents obtained in the presence of GSK. The decay in the current was fitted to a single exponential to obtain the time course of inhibition ($\tau$). Recovery from inhibition was determined by activating TRPV4 channels with GSK101 (300 nM), then adding salbutamol (500 μM) for 5 min, and then exposing the patches to GSK101 again. Data were obtained by averaging the results of several patches for each time point.

For single-channel recordings, borosilicate glass (30 MΩ) pipettes were used. Recordings were obtained at +60 mV by acquiring several traces of 1- to 3-s duration. The effect of salbutamol on single TRPV4 channels activated by GSK was studied in the outside-out configuration. Currents were low-pass-filtered at 3 kHz and sampled at 50 kHz. Patches containing only one channel activated by different compounds were identified as those that did not contain overlapping opening events.

The single-channel current (i) in each condition was determined by building all point histograms from traces with clear closings and openings. The resulting histograms were fitted to a Gaussian function, where the peak corresponded to single-channel openings and closures were identified with the half-threshold crossing technique, to compile a list of durations of all open events in a single sweep (Islas, 2015). Channel open probability (Po) was calculated as the sum of the total open time divided by the sweep duration. Recordings were performed in the absence of $Ca^{2+}$ to avoid block/desensitization of the channel.

### Mutagenesis

Mutations in the human TRPV4 channels were constructed by a two-step PCR method, as previously described (Salazar et al, 2008, 2009). Mutations in various regions of the TRPV4 channels were constructed using a method involving oligonucleotides synthesized to contain a mutation in combination with WT oligonucleotides in PCR amplifications of fragments of the complementary DNA. The product of the PCR was then cut with two different restriction enzymes to generate a cassette containing the mutation. The cassette was ligated into the channel complementary DNA cut with the same two restriction enzymes. The entire region of the amplified cassette was sequenced to confirm the mutation and ensure against second-site mutations.

### Computational modeling

To evaluate the interaction of salbutamol with the TRPV4 channel, molecular docking simulations were carried out. The structure of the human channel was built using *Xenopus tropicalis* TRPV4 as a template (≈85% of sequence identity). A total of 2,000 hTRPV4 models were built with the software Modeller 9.25 (Eswar et al, 2007). The best model was selected as that with the lowest Molpdf energy value of Modeller and the highest Procheck score (Laskowski et al, 1993). TRPV4 and salbutamol molecules were

prepared for docking with the software AutoDockTools 1.5.7 (Morris et al, 2009), assigning Gasteiger partial charges. Salbutamol is formulated as a racemic mixture of the R- and S-isomers (compound CID in PubChem is 2083). The R-isomer of salbutamol was used in this study because it has at least 100 times greater affinity for the ß2-receptor (Penn et al, 1996; Ameredes & Calhoun, 2009) than the S-isomer and the S-isomer has been associated with toxicity (Mitra et al, 1998; Volcheck et al, 2005). Search space for docking was initially demarcated through a grid box large enough to cover the entire surface of the channel. The grid parameters were generated with AutoGrid4.2.6 (Morris et al, 1998). The Lamarckian genetic algorithm of AutoDock4.2.6 (Morris et al, 1998, 2009) was used to model the TRPV4–salbutamol binding conformations. Docking refinement was performed, gradually reducing the conformational searching area. A total of 1,000 TRPV4 channel–salbutamol conformations were generated, which were clustered and analyzed with VMD 1.9.3 software (Humphrey et al, 1996).

## Statistical analysis

Group data are reported as the mean ± SD. The two-tailed $t$ test and one-way analysis of variance, followed by Tukey's post hoc test, were used for group comparison and calculated with Prism software (Dotmatics). Significant differences between means were considered to exist when the $P$-value was less than 0.01 or 0.05, as indicated.

# Supplementary Information

# Acknowledgements

We thank the following personnel from Instituto de Fisiología Celular, UNAM: Ana María Escalante Gonzalbo, Francisco Pérez Eugenio, Juan Manuel Barbosa Castillo, and Gerardo Coello Coutiño for technical assistance with computer software and hardware; Laura Kawasaki for help with technical assistance with experiments; and Ana E López-Romero for preliminary control experiments. We thank Drs. Sidney A Simon and Roberto Coria for helpful discussion of our article. M Benítez-Angeles is a doctoral student from the Programa de Doctorado en Ciencias Biomédicas, Universidad Nacional Autónoma de México (UNAM), and received a fellowship from the Consejo Nacional de Humanidades, Ciencias y Tecnologías (CONAHCyT; 1002182), and E Juárez-González received a fellowship 902482 from CONAHCyT. This work is in fulfillment of the requirements for a doctoral degree of the Programa de Doctorado en Ciencias Biomédicas for M. Benítez-Angeles at the Universidad Nacional Autónoma de México. This research was funded by the Dirección General de Asuntos del Personal Académico (DGAPA)-Programa de Apoyo a Proyectos de Investigación e Innovación Tecnológica (PAPIIT) (grant number IN200720 to T Rosenbaum and grant number IN215621 to LD Islas), and CONACyT (grant number A1-S-8760) and Secretaría de Educación, Ciencia, Tecnología e Innovación del Gobierno de la Ciudad de México (grant number SECTEI/208/2019) to T Rosenbaum. A Vergara-Jaque was funded by Fondo Nacional de Desarrollo Científico y Tecnológico (FONDECYT; grant number 1220110). The Millennium Nucleus of Ion Channel-Associated Diseases is a Millennium Nucleus supported by the National Agency of Research and Development (ANID), Chile.

## Author Contributions

M Benítez-Angeles: conceptualization, data curation, formal analysis, supervision, validation, investigation, methodology, and writing—original draft, review, and editing.
E Juárez-González: conceptualization, data curation, formal analysis, validation, investigation, methodology, and writing—original draft, review, and editing.
A Vergara-Jaque: conceptualization, data curation, software, formal analysis, supervision, funding acquisition, validation, investigation, visualization, methodology, and writing—original draft, review, and editing.
I Llorente: conceptualization, data curation, formal analysis, validation, methodology, and writing—original draft, review, and editing.
G Rangel-Yescas: conceptualization, data curation, formal analysis, validation, investigation, visualization, methodology, and writing—original draft, review, and editing.
SC Thébault: conceptualization, supervision, validation, investigation, methodology, and writing—original draft, review, and editing.
M Hiriart: conceptualization, supervision, validation, investigation, visualization, and writing—original draft, review, and editing.
LD Islas: data curation, formal analysis, supervision, funding acquisition, validation, investigation, visualization, methodology, and writing—original draft, review, and editing.
T Rosenbaum: conceptualization, resources, data curation, formal analysis, supervision, funding acquisition, validation, investigation, visualization, methodology, project administration, and writing—original draft, review, and editing.

## Conflict of Interest Statement

The authors declare that they have no conflict of interest.

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
