## [Reviewer comments · Life Science Alliance]

Life Science Alliance

Unconventional interactions of the TRPV4 ion channel with beta-adrenergic receptor ligands

Miguel Benítez-Angeles, Emmanuel Juárez-González, Ariela Vergara-Jaque, Itzel Llorente, Gisela Rangel-Yescas, Stephanie Thébault, Marcia Hiriart, Leon Islas, and Tamara Rosenbaum

DOI: <https://doi.org/10.26508/lsa.202201704>

Corresponding author(s): Tamara Rosenbaum, National Autonomous University of Mexico

Review Timeline:

Submission Date:	2022-09-02
Editorial Decision:	2022-10-05
Revision Received:	2022-11-14
Editorial Decision:	2022-12-02
Revision Received:	2022-12-05
Accepted:	2022-12-08

Scientific Editor: Novella Guidi

Transaction Report:

October 5, 2022

Re: Life Science Alliance manuscript #LSA-2022-01704-T

Dr. Tamara Rosenbaum
Universidad Nacional Autónoma de México
Instituto de Fisiología Celular
Ciudad Universitaria, Circuito exterior S/N
Mexico, D.F. 4510

Dear Dr. Rosenbaum,

Thank you for submitting your manuscript entitled "Unconventional interactions of the TRPV4 ion channel with beta-adrenergic receptor ligands" to Life Science Alliance. The manuscript was assessed by expert reviewers, whose comments are appended to this letter. We invite you to submit a revised manuscript addressing the Reviewer comments.

Thank you for this interesting contribution to Life Science Alliance. We are looking forward to receiving your revised manuscript.

Sincerely,

B. MANUSCRIPT ORGANIZATION AND FORMATTING:

Reviewer #1 (Comments to the Authors (Required)):

In this work by Benitez-Angeles, the inhibition of TRPV4 by the common asthma and COPD drug salbutamol is explored with electrophysiology and computational modeling. This is an excellent and straight-forward paper. The experiments are nicely controlled and well done. It is a classic ion channel/drug paper. I have only minor comments that could be considered by the authors to improve the paper. My specific comments are below:

1. Please discuss the likely cause of the 21.5% vehicle-only inhibition shown in Figure 1.
2. What is the concentration of salbutamol in an inhaler? Does the concentration-dependence of TRMPV4 inhibition (~81 microMolar) fall within the expected concentration of the drug commonly given to humans? Please discuss.
3. Please discuss the differences between R- and S-salbutamol in the text. Likewise, how much of each isomer you would expect in the solutions used in these experiments and how would this heterogeneity affect the presented experimental binding constants.
4. Did the authors do a computational search for both R- and S-isomer docking to TRPV4? Please discuss in the text.

Reviewer #2 (Comments to the Authors (Required)):

This is a carefully done solid study that describes the unconventional inhibition of polymodal TRPV4 ion channel by bronchodilator salbutamol and similar ligands of beta2-adrenergic receptors. Additionally, the authors pinned the site of interaction of these compounds with TRPV4. Using site-directed mutagenesis guided by molecular docking simulations the authors identified residues in the outer region of the channel that are responsible for interaction with salbutamol. The manuscript is solid, is well written and the data support the conclusions. This study is also physiologically important and provides important new insight into the regulation of this important molecule. I have only a few minor suggestions:

1. Since TRPV4 can be sensitive to extracellular pH, and salbutamol water solutions are usually kept at acidic pH, have the authors tested whether the addition of any of the compounds changes the pH of the bath solution?
2. Given micromolar concentrations of salbutamol required for TRPV4 inhibition (IC50 ~ 80uM), can this be in line with salbutamol concentrations in the patient's body during anti-asthmatic treatment? Such information could be helpful to add to the discussion.

Reviewer #3 (Comments to the Authors (Required)):

The submitted manuscript investigates the inhibition of the TRPV4 ion channel by the bronchodilator salbutamol. TRPV4 warrants investigation, as it is present in several different tissues and has been associated with various significant cellular functions. As a consequence, overexpression or increased activity can lead to a list of pathologies as highlighted by the authors. Although TRPV4 inhibitors have been described, they have limited safety profiles and there remain gaps in the knowledge of molecular mechanisms of action.

This manuscript reports that ligands of beta-adrenergic receptors including salbutamol inhibit TRPV4's activation. The manuscript also provides evidence for the mechanism of inhibition through closing of the channel by direct interaction, and shows that inhibition is specific to TRPV4 and not TRPV1. This is a comprehensive study presenting significant results from both multi-channel and single channel recordings of wild-type and mutant channels and computational modelling. The potential use of salbutamol offers the advantage of already being widely used as a bronchodilator, and the data show convincing partial inhibition. Although it is acknowledged by the authors that TRPV4 channels respond at higher concentrations than those to which b-AR's are sensitive. That said, the presented mechanistic data provide important insight into the molecular mechanisms that down regulate TRPV4 activity so will contribute to the continued development of novel therapeutic inhibitors for TRPV4. The manuscript is well written, including substantive and relevant background literature and clear aims of the study and discussion. Please see below for comments/queries for individual sections that need addressing:

Results and Figures

Salbutamol inhibits activation of TRPV4, but not of TRPV1

1. The authors do not state the percentage of inhibition after exposure to vehicle solution in the closed and open state experiments. Although significance values are shown in the Figure legends the actual percentage values must also be stated in the text and/or Figure legend.
2. Authors state in the main text that 'most' electrophysiological experiments were carried out in the absence of Ca²⁺. It needs to be made more clear which experiments were in the presence or absence of Ca²⁺ and the physiological interpretation of results obtained when Ca²⁺ is not present. In the Figure legend for Figure 1, it states + Ca in the 'open state' experiments. If this is the case it needs to be labelled as so in Figure 1E
3. Figure 1: There needs to be a Figure legend added to indicate what the orange represents for TRPV1.
4. In Figure 2B, do the authors have any further time points after 300 seconds as from the graph, the inhibition has not quite plateaued at the time points shown. From this graph it looks possible that the authors may be underestimating the maximal inhibition time course.
5. Page 9, first sentence - should read 'fully activate'

Effects of short- and long-acting bronchodilators on TRPV4 activation

6. Were dose response curves performed for the short and long acting bronchodilators? If not, how was the concentration used determined - has this been done elsewhere? If so a reference should be added.
7. Was the inhibition in any of the compounds reversible? Did the authors test any of them on TRPV1?
Antagonism of b-adrenoreceptors also results in TRPV4 inhibition.
- 8 Second paragraph page 11 - 'since salbutamol 500 μ M inhibits TRPV4 currents in about a 50%' this sentence is incomplete.
- 9 Labels are needed on Figure 6 traces and graphs - purple and blue must be labelled.
- 10 There needs to be consistency with the axis labelling of the average data. The y axis changes in Figure 8 and EV1 -3.
- 11 In Figure EV2, it would be helpful to have the WT trace shown. Also, when looking at T689A in A, it looks like there is a reduction in inhibition. I would suggest that a more representative trace is shown.
- 12 Data shown in Figure EV3 although mentioned in the discussion, is not described in the results section - this must be added.

Reviewer #1

In this work by Benitez-Angeles, the inhibition of TRPV4 by the common asthma and COPD drug salbutamol is explored with electrophysiology and computational modeling. This is an excellent and straight-forward paper. The experiments are nicely controlled and well done. It is a classic ion channel/drug paper. I have only minor comments that could be considered by the authors to improve the paper. My specific comments are below:

We thank the reviewer for useful comments that have improved our manuscript. Please find below our responses to your suggestions.

1. Please discuss the likely cause of the 21.5% vehicle-only inhibition shown in Figure 1E

Several TRP channels have been shown to depend upon the presence of intracellular factors, which when removed (as what happens in excised membrane patches like the ones used in our study), can produce ion channel rundown. TRPV4 has been previously shown to exhibit some PIP2-dependent rundown ([10.1073/pnas.1220231110](https://doi.org/10.1073/pnas.1220231110)) but other intracellular effectors (please see lines 369-380), not described to date, may be important to maintain its activity.

Importantly, please note that the decrease in current in the presence of the effective bronchodilators, is statistically different ($p < 0.0001$ for closed state and $p = 0.0134$ for open state) from that observed only with the vehicle (please see lines 154-157 and 1006-1014). We thank you for this observation and we have included a sentence to explain this point in the manuscript,

2. What is the concentration of salbutamol in an inhaler? Does the concentration-dependence of TRPV4 inhibition (~81 microMolar) fall within the expected concentration of the drug commonly given to humans? Please discuss.

An inhaler typically has between 100-150 $\mu\text{M}/\text{kg}$ salbutamol per dose (and can be used 3 to 4 times a day (with a lifetime of 3-6 h; [10.1007/164_2016_64](https://doi.org/10.1007/164_2016_64)). Interestingly, nebulized salbutamol has been found to be more effective, as compared to its systemic administration. A possible explanation for this is that there appears to be no direct biotransformation of salbutamol in the lungs (with a half-life between 2–7 h), when applied with an inhaler. We have now discussed this in the manuscript (please see lines 184-189): “It is interesting to note that an inhaler, which can be used 3 or 4 times a day, typically contains between 100-150 $\mu\text{M}/\text{kg}$ salbutamol per dose. Interestingly, nebulized salbutamol has been found to be more effective, as compared to its systemic administration. A possible explanation for this is that there appears to be no direct biotransformation of salbutamol in the lungs (with a half-life between 2–7 h), when applied with an inhaler (Gad, 2014).”

80 μM of salbutamol is equivalent to 19 $\mu\text{g}/\text{mL}$ (molecular weight = 239.311 g/mol). Several studies reported plasma levels of salbutamol after its intake (inhaled or oral) in the treatment of acute asthma between 1.75 to 18.77 ng/mL in adults (10.1249/MSS.0b013e3181b2e87d) and 2.77 to 18.22 ng/mL in infants (10.1007/s00228-010-0787-4). The micromolar concentrations that we found to inhibit TRPV4 seem therefore to be 1000 times higher than patient's concentrations during asthmatic treatment, suggesting that at therapeutic doses, salbutamol may not inhibit TRPV4 in the lung. It may however be noticed that urine levels of salbutamol during asthma treatment rise up to 2422.2 ng/mL, indicating that some tissues may receive doses of salbutamol at which TRPV4 is inhibited. Please see lines 425-432.

3. Please discuss the differences between R- and S-salbutamol in the text. Likewise, how much of each isomer you would expect in the solutions used in these experiments and how would this heterogeneity affect the presented experimental binding constants.

Salbutamol is a chiral compound usually found as a 50:50 racemic mixture of two enantiomers (stereoisomers), which are molecules with non-superimposable mirror images. It has one chiral center in the ethanolamine tail, resulting in the two enantiomers (S)- and (R)- for the hydroxyl-group position.

The first experiments were performed using the racemic mixture of salbutamol, hence, in our preliminary conclusions on the partial inhibition obtained with salbutamol (Figures 1 and 2), we hypothesized that only one fraction was responsible for the inhibitory effects. However, when we tested R-salbutamol (levalbuterol, please see Figure 3), we obtained the same result, with a similar magnitude of inhibition as the one observed with the racemic mixture. Note that S-salbutamol alone is not clinically used since it has been reported to either not have any kind of effect on asthmatic patients or to produce bronchoconstriction. We have added a few sentences related to this to the discussion section (please see lines 364-372 and answer to point #4). Thank you for this observation.

4. Did the authors do a computational search for both R- and S-isomer docking to TRPV4? Please discuss in the text.

We only performed the computational search for the R-isomer docking to TRPV4 for the following reasons: As stated, salbutamol is formulated as a racemic mixture of the R- and S-isomers. The R-isomer has 150 times greater affinity for the β_2 -receptor than the S-isomer, and the S-isomer has been associated with toxicity, which led to the development of levalbuterol, but the high cost of this pure enantiomer has deterred its widespread clinical use. Thank you, we have clarified this in lines 574-584 in the methods section and 364-380 in the discussion section.

These paragraphs state that:

“Salbutamol is formulated as a racemic mixture of the R- and S-isomers (compound CID in Pubchem is 2083). The R-isomer of salbutamol was used in this study because it has at least 100 times greater affinity for the β 2-receptor (Penn et al., 1996; Ameredes and Calhoun, 2009) than the S-isomer and the S-isomer has been associated with toxicity (Mitra et al., 1998; Volcheck et al., 2005).”

And:

“Salbutamol is a chiral compound usually found as a 50:50 racemic mixture of two enantiomers (stereoisomers), which are molecules with non-superimposable mirror images. It has one chiral center in the ethanolamine tail, resulting in the two enantiomers (S)- and (R)- for the hydroxyl-group position. Our experiments using the racemic mixture of salbutamol, show partial inhibition of TRPV4 (Figures 1 and 2) and the R-salbutamol enantiomer (levalbuterol, Figure 3) produces a similar inhibitory effect to that observed with the racemic mixture. We do not know whether the S- enantiomer inhibits TRPV4, but it noteworthy that S-salbutamol alone is not clinically used since it has been reported to either not have any kind of effect on asthmatic patients or to produce bronchoconstriction (Templeton et al., 1998; Gumbhir-Shah et al., 1999). It is also important to mention that most of the experiments in this study were performed in the absence of Ca^{2+} because, as stated above, this ion can produce inhibition of TRPV4. In this sense, it is hard to hypothesize how the compounds here tested would affect the function of TRPV4 under physiological conditions, since several scenarios are viable with the channel possibly exhibiting smaller currents because of the presence of Ca^{2+} , but also being subject to potentiation of its activity by changes in its phosphorylation state (Wegierski et al., 2009), changes in PIP2 concentration (Garcia-Elias et al., 2013; Harraz et al., 2018) and other changes in the cellular environment.”

Reviewer #2

We thank the reviewer for thoughtful comments that have improved our manuscript. Please find below our responses to your suggestions.

This is a carefully done solid study that describes the unconventional inhibition of polymodal TRPV4 ion channel by bronchodilator salbutamol and similar ligands of beta2-

adrenergic receptors. Additionally, the authors pinned the site of interaction of these compounds with TRPV4. Using site-directed mutagenesis guided by molecular docking simulations the authors identified residues in the outer region of the channel that are responsible for interaction with salbutamol. The manuscript is solid, is well written and the data support the conclusions. This study is also physiologically important and provides important new insight into the regulation of this important molecule. I have only a few minor suggestions:

1. Since TRPV4 can be sensitive to extracellular pH, and salbutamol water solutions are usually kept at acidic pH, have the authors tested whether the addition of any of the compounds changes the pH of the bath solution?

This is a good question, because TRP channels are indeed sensitive to pH changes. For electrophysiological experiments, salbutamol was freshly prepared every day before use by weighing it and dissolving in water and then, immediately adding it to the recording solutions, buffered with HEPES, at the desired concentration. We did in fact check the final pH of our recording solution with salbutamol and even at the highest concentration used (which was 1 mM), the pH was not significantly changed and remained at 7.2 with 3 mM HEPES to maintain the pH. We have clarified this in lines 502-504 in the manuscript: “pH was measured after addition of compounds to ensure that it was not modified in the recording buffered solutions to which membrane patches containing TRPV4 channels were exposed.”

2. Given micromolar concentrations of salbutamol required for TRPV4 inhibition ($IC_{50} \sim 80\mu M$), can this be in line with salbutamol concentrations in the patient's body during anti-asthmatic treatment? Such information could be helpful to add to the discussion.

Thank you for this suggestion. We have added lines 425-432 in the manuscript: “80 μM of salbutamol is equivalent to 19 $\mu g/mL$ (salbutamol molecular weight = 239.311 g/mol). Several studies have reported plasma levels of salbutamol after its intake (inhaled or oral) in the treatment of acute asthma that range between 1.75 and 18.77 ng/mL in adults (Elers et al., 2010) and 2.77 to 18.22 ng/mL in infants (Rotta et al., 2010) but, in lung edema fluid from patients treated with salbutamol, the mean concentration reaches 700 ng/ml (Atabai et al., 2002). Moreover, levels of salbutamol in the urine during asthma treatment were shown to increase up to 2422.2 ng/mL (Elers et al., 2010). Thus, the micromolar concentrations that we found inhibit TRPV4 suggest that some tissues receive doses of salbutamol at which TRPV4 is inhibited.”

Reviewer #3 (Comments to the Authors (Required)):

The submitted manuscript investigates the inhibition of the TRPV4 ion channel by the bronchodilator salbutamol. TRPV4 warrants investigation, as it is present in several different tissues and has been associated with various significant cellular functions. As a consequence, overexpression or increased activity can lead to a list of pathologies as highlighted by the authors. Although TRPV4 inhibitors have been described, they have limited safety profiles and there remain gaps in the knowledge of molecular mechanisms of action.

This manuscript reports that ligands of beta-adrenergic receptors including salbutamol inhibit TRPV4's activation. The manuscript also provides evidence for the mechanism of inhibition through closing of the channel by direct interaction, and shows that inhibition is specific to TRPV4 and not TRPV1. This is a comprehensive study presenting significant results from both multi-channel and single channel recordings of wild-type and mutant channels and computational modelling. The potential use of salbutamol offers the advantage of already being widely used as a bronchodilator, and the data show convincing partial inhibition. Although it is acknowledged by the authors that TRPV4 channels respond at higher concentrations than those to which b-AR's are sensitive. That said, the presented mechanistic data provide important insight into the molecular mechanisms that down regulate TRPV4 activity so will contribute to the continued development of novel therapeutic inhibitors for TRPV4.

The manuscript is well written, including substantive and relevant background literature and clear aims of the study and discussion. Please see below for comments/queries for individual sections that need addressing:

We thank the reviewer for comments and suggestions that have enriched our manuscript. Please find below our answers to your concerns.

Results and Figures

Salbutamol inhibits activation of TRPV4, but not of TRPV1

1. The authors do not state the percentage of inhibition after exposure to vehicle solution in the closed and open state experiments. Although significance values are shown in the Figure legends the actual percentage values must also be stated in the text and/or Figure legend.

Thank you for this observation. The value for percentage of inhibition (rundown) by the vehicle for the duration of the experiment is stated in lines 154-157 in the text and in lines 991-998 (also please see below) in the legend of Figure 1:

“When membrane patches were exposed to vehicle only (TRIS), there was only a $21.2 \pm 9.7\%$ decrease (**Fig. 1E**, +120 mV, empty triangles) in currents, as compared to initial currents obtained in the presence of GSK and could be due to the removal of an intracellular factor required for sustained activity upon membrane patch excision (Garcia-Elias et al., 2013).” Please see response # 1 to Reviewer 1.

Moreover, we have explained that there is rundown of the channel in the open state (in the presence of GSK and vehicle) and stated the values in lines 166-172 of the text and in figure legend 1: “Application of vehicle only in the closed state produced $21.2 \pm 9.7\%$ (empty triangles, $n = 13$) of the inhibition and $16.16 \pm 8.2\%$ (empty ties, $n= 15$) when vehicle was applied in the presence of GSK101. In the case of TRPV1, 2.6 ± 16.8 ($n = 10$) of currents were inhibited after treatment with salbutamol (orange squares).” Please see lines 1006-1014.

2. Authors state in the main text that 'most' electrophysiological experiments were carried out in the absence of Ca²⁺. It needs to be made more clear which experiments were in the presence or absence of Ca²⁺ and the physiological interpretation of results obtained when Ca²⁺ is not present. In the Figure legend for Figure 1, it states + Ca in the 'open state' experiments. If this is the case it needs to be labelled as so in Figure 1E

Thank you for this suggestion. The manuscript now clearly states that all experiments except those in Figure 1E were performed in the absence of Ca²⁺ (please see lines 151-154 and in the Methods section lines 478-481). We have also added the following text to the discussion section (please see lines 372-380): “It is also important to mention that most of the experiments in this study were performed in the absence of Ca²⁺ because, as stated above, this ion can produce inhibition of TRPV4. In this sense, it is hard to hypothesize how the compounds here tested would affect the function of TRPV4 under physiological conditions, since several scenarios are viable with the channel possibly exhibiting smaller currents because of the presence of Ca²⁺, but also being subject to potentiation of its activity by changes in its phosphorylation state (Wegierski et al., 2009), changes in PIP2 concentration (Garcia-Elias et al., 2013; Harraz et al., 2018) and other changes in the cellular environment.” Indeed, further studies are needed to clarify this issue.

We have added the legend “open state” to Figure 1E. Thank you for pointing this out.

3. Figure 1: There needs to be a Figure legend added to indicate what the orange represents for TRPV1.

Done. Thank you.

4. In Figure 2B, do the authors have any further time points after 300 seconds as from the graph, the inhibition has not quite plateaued at the time points shown. From this graph it looks possible that the authors may be underestimating the maximal inhibition time course. We have added a few more points to the time course in Figure 2B. Inhibition is not significantly changed after 5 mins in the presence of the compound. We have added two more points to Figure 2B and explained in the text that, in the presence of vehicle only, at 7 mins, current decay is already around 50%. Please see lines 192-195.

5. Page 9, first sentence - should read 'fully activate'

We have fixed this. Please see line 208.

Effects of short- and long-acting bronchodilators on TRPV4 activation

6. Were dose response curves performed for the short and long acting bronchodilators? If not, how was the concentration used determined - has this been done elsewhere? If so a reference should be added.

No other study has shown inhibition of TRPV4 by the compounds tested here. We chose a concentration similar to the one where maximal current inhibition is attained in the presence of salbutamol (500 μ M) and, as shown in Figure 3, addition of each compound at this concentration did not produce more (or less) current inhibition in comparison with salbutamol. We have now stated this more clearly in the manuscript (please see lines 217-225). We do not expect that we would learn much more from performing the full dose-responses for each of the other compounds in terms of the present study.

7. Was the inhibition in any of the compounds reversible? Did the authors test any of them on TRPV1?

This is a good question. We have performed new experiments and we now show in Figure 2C the recovery from inhibition of TRPV4 by salbutamol. These data show that channels do not recover from this inhibition after 10 mins. It is possible that this result is a combination of the channel not recovering from inhibition and from its function running down after excision of the membrane patch. Please see lines 195-197 in the Results section, lines 537-539 in the Methods section and 1023-1025 in Figure 2 legend.

As for the effects of compounds tested here on TRPV1, we had tested salbutamol (Figure 1) and we have also tested terbutaline and isoprenaline on TRPV1, and there is no significant effect of

these compounds on this channel (please see below for Reviewer Figure 1). For a future study, we plan on testing several more compounds on TRPV1 and other TRP channels.

Antagonism of β -adrenoreceptors also results in TRPV4 inhibition.

8 Second paragraph page 11 - 'since salbutamol 500 μ M inhibits TRPV4 currents in about a 50%' this sentence is incomplete.

Thank you. We rewrote the paragraph and it now reads: "To further substantiate this conclusion, we performed experiments where we incubated TRPV4-expressing HEK293 cells for 24 h with 11 μ M ICI and measured GSK-induced TRPV4 currents from excised HEK293 cell membrane patches in the presence of a concentration near the K_D of salbutamol (100 μ M), which is expected to inhibit around 25% of the total currents since the maximal concentration of 500 μ M salbutamol inhibits 50% of these currents. **Fig. 5D and E** showed that antagonism of β_2 -AR's did not affect inhibition of TRPV4 by salbutamol ($23.6 \pm 17.3\%$ with salbutamol vs. $23.4 \pm 18.9\%$ with ICI preincubation). These results are in accordance with what we would expect if salbutamol was acting directly on TRPV4 and independently of β_2 -AR-associated signaling pathways." Please see lines 268-273.

9 Labels are needed on Figure 6 traces and graphs - purple and blue must be labelled.

Thank you. We have fixed this.

10 There needs to be consistency with the axis labelling of the average data. The y axis changes in Figure 8 and EV1 -3.

Thank you. We fixed this.

11 In Figure EV2, it would be helpful to have the WT trace shown. Also, when looking at T689A in A, it looks like there is a reduction in inhibition. I would suggest that a more representative trace is shown.

Thank you. We have now added a trace for WT TRPV4 and replaced the one for T689A by one which is more representative of the average in Figure EV2 (now Figure S2).

12 Data shown in Figure EV3 although mentioned in the discussion, is not described in the results section - this must be added.

Thank you. We have added lines 318-322 discussing Figure EV3 (now Figure S3) in the results section.

[Figure has been removed by *LSA* Editorial Staff per authors' request.]

December 2, 2022

RE: Life Science Alliance Manuscript #LSA-2022-01704-TR

Dr. Tamara Rosenbaum
National Autonomous University of Mexico
Instituto de Fisiología Celular
Ciudad Universitaria, Circuito exterior S/N
Mexico, D.F. 4510
Mexico

Dear Dr. Rosenbaum,

Thank you for submitting your revised manuscript entitled "Unconventional interactions of the TRPV4 ion channel with beta-adrenergic receptor ligands". We would be happy to publish your paper in Life Science Alliance pending final revisions necessary to meet our formatting guidelines.

A. FINAL FILES:

B. MANUSCRIPT ORGANIZATION AND FORMATTING:

****The license to publish form must be signed before your manuscript can be sent to production. A link to the electronic license to**

publish form will be sent to the corresponding author only. Please take a moment to check your funder requirements.**

Sincerely,

Reviewer #3 (Comments to the Authors (Required)):

The authors have thoroughly addressed the points I raised and have amended the manuscript accordingly. I am happy to support the publication in it's amended form.

December 8, 2022

RE: Life Science Alliance Manuscript #LSA-2022-01704-TRR

Dr. Tamara Rosenbaum
National Autonomous University of Mexico
Instituto de Fisiología Celular
Ciudad Universitaria, Circuito exterior S/N
Mexico, D.F. 4510

Dear Dr. Rosenbaum,

Thank you for submitting your Research Article entitled "Unconventional interactions of the TRPV4 ion channel with beta-adrenergic receptor ligands". It is a pleasure to let you know that your manuscript is now accepted for publication in Life Science Alliance. Congratulations on this interesting work.

DISTRIBUTION OF MATERIALS:

Again, congratulations on a very nice paper. I hope you found the review process to be constructive and are pleased with how the manuscript was handled editorially. We look forward to future exciting submissions from your lab.

Sincerely,
